# Factors associated with mortality due to neonatal pneumonia in India: a protocol for systematic review and planned meta-analysis

N Sreekumaran Nair,[1] Leslie Edward Lewis,[2] Theophilus Lakiang,[1] Myron Godinho,[1] Shruti Murthy,[1] Bhumika T Venkatesh[1]

[1]Public Health Evidence South Asia (PHESA),Manipal University, Manipal, India
[2]Department of Pediatrics and Neonatology, Kasturba Medical College,Manipal University, Manipal, India

**Correspondence to**
Dr N Sreekumaran Nair; nsknairmanipal@gmail.com

## ABSTRACT

**Introduction** India contributes to the highest number of neonatal deaths globally. It also has the greatest number of pneumonia-related neonatal deaths in the developing world. We aim to systematically review the evidence for the factors associated with mortality due to neonatal pneumonia in the Indian context, to address the lack of consolidated evidence on this important issue.

**Methods and analysis** This protocol is part of a series of three reviews on neonatal pneumonia in India. Observational studies reporting on outcome of neonatal pneumonia in the Indian context, and published in English in peer-reviewed and indexed journals will be eligible for inclusion. Outcomes of this review will be the factors determining mortality due to neonatal pneumonia. A total of nine databases will be searched. Electronic and hand searching of published and grey literature will be performed. Selection of studies will be done in title, abstract and full text screening stages. Risk of bias, independently assessed by two authors, will be evaluated. Meta-analysis will be performed and heterogeneity assessed. Pooled effect estimates will be stated with 95% confidence intervals. Narrative synthesis will be done where meta-analysis cannot be performed. Publication bias will be evaluated and sensitivity analysis performed according to study quality. Quality of this review will be evaluated using AMSTAR (Assessing the Methodological quality of Systematic Reviews) and GRADE (Grades of Recommendation, Assessment, Development & Evaluation). A summary of findings table will be reported using GRADEPro.

**Ethics and dissemination** Since this is a review involving analysis of secondary data which is available in the public domain, and does not involve human participants, ethical approval was not required. The findings of the study will be shared with all stakeholders of this research. Knowledge dissemination workshops will be conducted with relevant stakeholders to transfer the evidence, tailored to the stakeholder (eg, policy briefs, publications, information booklets, etc).

## Strengths and limitations of this study

► Most documented literature on mortality predictors in pneumonia address post-neonatal age groups. First review to consolidate and assess research on predictors of mortality due to pneumonia among neonates in the Indian context.

► A comprehensive search strategy was developed for nine databases including relevant regional databases and grey literature.

► We considered only English language studies and studies which assessed pneumonia independent of neonatal sepsis. We attempted to capture definition of neonatal pneumonia reported in each individual study.

## INTRODUCTION

Pneumonia is the single largest cause of death in children worldwide accounting for almost one fifth of the under-five child deaths.[1] Ninety percent of these occur in developing countries; mostly in South Asia and Sub-Saharan Africa.[2][3] India accounts for the greatest number of childhood pneumonia deaths among high burden countries.[3] India also contributes to a greater proportion of global neonatal deaths than any other country,[4] and neonatal mortality comprises over half of national under-five deaths.[5] In India, pneumonia accounts for 16% of neonatal deaths, compared with just 3% of global neonatal deaths, highlighting it as a serious cause for concern.[6] With 57% of India's under-five deaths occurring in the neonatal period and high burden of pneumonia, effectively tackling neonatal pneumonia is important in controlling the national and regional neonatal mortality rate.[7] However, there remains a lack of consolidated research for factors responsible for mortality in neonatal pneumonia, especially from the Indian context. This would provide the evidence required to inform decision-making in neonatal health, both for policy-making and programme implementation. Our objective is to systematically review and

assess the evidence for factors associated with mortality among neonates with pneumonia in the Indian context.

This protocol is part of a larger mixed-methods research project consisting of a qualitative study and a trilogy of reviews on neonatal pneumonia in India addressing

- ► factors associated with neonatal pneumonia,
- ► treatment options and barriers to the case management of neonatal pneumonia
- ► factors associated with mortality due to neonatal pneumonia.

## METHODS AND ANALYSIS

This protocol has been developed according to the 'Meta-analysis of Observational Studies in Epidemiology' (MOOSE) guidelines and Preferred Reporting Items for Systematic reviews and Meta-Analysis- Protocol' (PRISMA-P) guidelines.[8 9] This review will be conducted from August 2016 to October 2017.

## CRITERIA FOR CONSIDERING STUDIES FOR THIS REVIEW
### Types of studies
#### Inclusion criteria
Published studies conducted on neonates with pneumonia in the Indian context (irrespective of the diagnostic criteria used) will be eligible for inclusion. Studies should have been published in English in indexed and peer-reviewed journals between 1980- April 2017. Eligible study designs include (a) analytic study designs (case-control studies, cohort studies, analytical cross sectional studies) and (b) descriptive studies (case report, case series, cross-sectional studies), reports of secondary data analyses, outcome studies, and fact sheets which report a quantitative analysis of factors associated with mortality in neonatal pneumonia.

#### Exclusion criteria
Studies will be excluded if they are letters, editorials, commentaries, reviews, meta-analysis, qualitative research, conference papers, reports which do not include a quantitative analysis of factors associated with mortality in neonatal pneumonia.

Type of participants: Neonates with pneumonia in the Indian context.

Outcome of interest of this review: Outcome of this review will be factors associated with mortality in neonatal pneumonia. Mortality from neonatal pneumonia had to occur within the neonatal period. In this context, factors are defined as any attribute, characteristic or exposure of an individual that increases the likelihood of mortality due to neonatal pneumonia[10]; these may be related to patient, parent, maternal and pregnancy, environment, health system, iatrogenic or any other aspects. The list is not exhaustive and will be modified based on the evidence compiled from the systematic review. Definitions as reported by the authors will be considered and captured in the review.

## Search methods for identification of studies
An appropriate and comprehensive search strategy with relevant search terms for all data sources mentioned will be developed and pilot tested before final search.

Electronic searches: We will search PubMed, EMBASE, Web of Science, SCOPUS, CINAHL, Ovid MEDLINE, ProQuest, WHO IMSEAR (WHO Index Medicus South East Asian Region) and IndMED.

Hand searching: Hand searching will be conducted for (a) journal volumes which are not included in electronic databases.

Searching the grey literature: Potential sources of grey literature will include Shodhganga (INFLIBNET) and Government of India databases.

Reference lists: Snowballing will be performed to screen the references of identified literature for potentially relevant studies. Additionally, experts, authors, researchers and relevant organisations of identified studies will be contacted to suggest other existing relevant studies.

An example of our search strategy for PubMED is presented in table 1.

## Data collection and management
The results (titles and/or abstracts) of the search will be managed using Endnote (v. x7). Study selection will be performed on Endnote (v. x7). Data will be extracted on Microsoft Excel 2007. Statistical analysis will be performed using STATA (v.13).

## Selection of studies
Studies will be reviewed based on the exclusion and inclusion criteria, by two authors independently (TL and MG) in three stages. During the first stage of title screening, titles of the studies identified from the search will be assessed for inclusion. If both authors reject a title, it will be excluded from the review. Titles approved by either author will move to abstract screening. In the next stage of abstract screening, abstracts of these selected titles that are approved by either author will be included for the final stage of full text screening. If both authors reject a study at this stage, it will be excluded from the review. In the third stage of full text screening, full texts of abstracts selected in the previous stage will be screened for eligibility. Only those studies approved by both authors will be included in the review. In the event of any disagreements, a third author (SM) and senior review authors (SN and LL) will arbitrate and a consensus will be reached on the inclusion of the study. Rationale for exclusion will be provided for all studies which get excluded through this process. A final list of articles will be prepared for data extraction. A PRISMA chart will be created, to outline and summarise this study selection process.[11]

## DATA EXTRACTION
The data extraction form was developed through the collaboration of authors, with necessary support from senior reviewers, subject and clinical experts, and

**Table 1**  Search strategy (PubMED)

**Strategy: #1 AND #2 AND #3 AND #4**

| #1 | ((((Neonate* OR childhood OR neonatal* OR newborn* OR 'young infant' OR child OR paediatric* OR 'neonatal period' OR infant* OR 'newborn infant'))) |
|---|---|
| #2 | ((((((((((((((((((((((((Pneumonia*) OR Pneumon*) OR 'community acquired pneumonia') OR 'congenital pneumonia') OR 'hospital acquired pneumonia') OR 'nosocomial pneumonia') OR 'ventilator associated pneumonia') OR 'early onset pneumonia') OR 'late onset pneumonia') OR 'infective pneumonia') OR 'infectious pneumonia') OR 'meconium aspiration syndrome') OR 'meconium aspiration') OR 'lipoid pneumonia') OR sepsis*) OR 'acute respiratory infections') OR 'early onset sepsis') OR 'chemical pneumonia') OR 'aspiration pneumonia') OR 'late onset sepsis') OR infection*) OR 'nosocomial infection') OR 'early onset infection') OR 'late onset infection') OR 'acute lower respiratory infection') OR 'hospital acquired infection') OR 'congenital infection') OR 'viral pneumonia') OR 'gastro esophageal reflux disease') OR 'cystic fibrosis') |
| #3 | ((Mortality* OR death* OR fatal* OR 'case fatality' OR 'case fatality rate'))) |
| #4 | (((('Risk factor' OR determinant* OR risk* OR predictor* OR 'relative risk' OR 'OR' OR 'attributable risk' OR 'population attributable fraction'))))) |

Geographical filter: India.
Language filter: English.
Period of publication: 1 January 1986- 1 August 2016.

statisticians. The form has been pilot-tested on one study of each type to ensure that it adequately facilitated the collection of all necessary information required for an effective meta-analysis. Broad categories under which data will be extracted include (a) Study Characteristics (b) Methodological characteristics (c) Factors identified (factors, type of data, measure of association calculated) and (d) Other important information. From the selected studies, data will be extracted with the use of a standardised, pre-tested data extraction form by two authors (SM and MG) independently. Any disagreements will be resolved by discussion and consensus between the authors.

### DEALING WITH MISSING DATA

In case of inadequacy, missing information, lack of clarity on information in methodology or outcomes are missing, authors of the respective studies will be contacted in an attempt to obtain the required details. A maximum of two email attempts will be made. Despite this, if the missing data retrieval is not possible, the study will be included in the systematic review and discussed in the narrative summary. However, this study will be excluded from meta-analysis and analysis of only the available data will be performed.

### ASSESSMENT OF RISK OF BIAS IN INCLUDED STUDIES

Risk of bias assessment will be done at the study level. Two authors (TL and MG) will independently assess the risk of bias at both the study and outcome level in included studies. Disagreements between the review authors over the risk of bias in particular studies will be resolved by discussion, with involvement of a third (SM) and senior authors (SN and LL) where necessary. The outcome of this appraisal will be discussed in the final narrative synopsis, where its implications on the outcome

of the meta-analysis will be discussed. The risk of bias of case-control and cohort studies will be assessed by using the Newcastle-Ottawa Scale (NOS).[12] An adapted NOS will be used for the appraisal of cross-sectional studies.[13 14] The quality of case series will be assessed using the Institute of Health Economics (IHE) criteria.[15]

### DATA ANALYSIS
### Data synthesis
A meta-analysis will be used to consolidate quantitative data. Data, as reported in the studies, will first be extracted. Next, if required, they will be transformed.

#### Categorical data
Categorical data will be summarised using measures such as OR and relative risk (RR) for risk estimation.

#### Continuous data
Continuous data will be summarised using the standardised mean difference (SMD).

The summary measures will be pooled according to the study design. Pooled effect estimates will be stated with 95% confidence intervals quantitatively and illustrated in a forest plot (using a logarithmic scale to present ratios in case event data is not available for analysis) along with tables where necessary.[16] We will summarise the characteristics and results of included studies using additional tables, supplemented by a narrative summary that will compare and evaluate methods used and principal results between studies. Further, we will describe factors for which a meta-analysis is impossible and the reasons for exclusion from meta-analysis will be provided.

Investigation of heterogeneity: A fixed-effects or a random-effects model will be employed, based on study heterogeneity, determined by calculating an $I^2$ statistic with 95% confidence intervals. Possible reasons for heterogeneity will be discussed.

Sub group analysis: Subgroup analysis will be done, when data are available, to compare risk factors according to, but not limited to (a) study design (b) type of neonatal pneumonia, (c) setting, and (d) timing of onset (early and late) of neonatal pneumonia. The final sub grouping will be decided after data extraction.

Meta-regression: Meta-regression will also be conducted in order to determine the effect of the covariates on the pooled effect size, and allow for adjustment of this effect to account for heterogeneity due to varying contexts and populations. The change in the pooled effect result will be measured to determine the role of the relevant study

Sensitivity analysis: Sensitivity analysis will be provided based on study quality.

Assessment of reporting bias: Publication bias will be assessed when there are more than ten studies included in the review by generating a funnel plot and performing Egger's test to assess the degree of asymmetry.

## QUALITY CONTROL OF THE SYSTEMATIC REVIEW AND META-ANALYSIS

The methodological quality of the systematic review will be evaluated using the 'A Measurement Tool to Assess Systematic Reviews' (AMSTAR) criteria.[17] The 'Grading of Recommendations Assessment, Development and Evaluation' (GRADE) assessment for the quality of evidence produced by systematic reviews, and a summary of findings table will be reported using GRADEPro.

## ETHICS AND DISSEMINATION
### Ethics

Since this is a review involving analysis of secondary data which is available in the public domain, and does not involve human participants, ethical approval was not required.

### Dissemination

The findings of the study will be shared with all stakeholders of this research. Knowledge dissemination workshops will be conducted with relevant stakeholders to transfer the evidence, tailored to the stakeholder (eg, policy briefs, publications, information booklets, etc).

### Reporting of the systematic review and meta-analysis

The findings of this systematic review will be reported in accordance with the PRISMA Guidelines and the MOOSE Guidelines.[8 11]

**Acknowledgements** We would like to thank the following individuals for their continuous support and guidance during this process of protocol development: Dr. Manoj Das, Director Projects, The INCLEN Trust International, New Delhi; Dr Anju Sinha, Deputy Director General, Scientist 'E', Division of Child Health, Indian Council of Medical Research, New Delhi; Dr. K.K. Diwakar, Professor and Head, Department of Neonatology, Associate Dean, Malankara Orthodox Syrian Church Medical College, Kerala; Mrs. Ratheebhai V., Senior Librarian and Information Scientist, at Manipal School at Communication, Manipal University, Manipal; Dr. Ravinder M. Pandey, Professor and Head, Department of Biostatistics, All India Institute of Medical Sciences, New Delhi; Dr. B. Shantharam Baliga, Professor, Department of Paediatrics, Kasturba medical college, Mangalore, Karnataka; Dr. Shirish Darak, Senior researcher, PRAYAS, Pune, Maharashtra; Dr. Unnikrishnan B., Associate Dean and Professor, Department of Community Medicine, Kasturba Medical College, Mangalore. We also thank Public Health Evidence South Asia (PHESA) and Manipal University, Manipal for providing the necessary institutional and infrastructural support for the project. We would also like to thank The INCLEN Trust International, New Delhi, and The Bill and Melinda Gates Foundation for the financial support which made this project possible.

**Contributors** SN is the guarantor of the review. SN, BV and LL conceived the research idea, provided overall technical guidance and reviewed the protocol. In addition, LL assisted in developing search terms. TL, MG and SM designed the protocol, drafted the manuscript, developed and pilot tested the search strategies and data extraction form.

**Funding** This project is supported by a grant from Bill and Melinda Gates Foundation (grantOPP1084307) to The INCLEN Trust International and sub-grant to Manipal University(subgrant INC2015GNT004). The views expressed through this project do not necessarilyrepresent the views of Bill and Melinda Gates Foundation or The INCLEN Trust Internationalor Manipal University.

**Competing interests** All authors have completed the ICMJE uniform disclosure form at www.icmje.org/coi_disclosure.pdf and declare: all authors had financial support (grants) from Bill and Melinda Gates Foundation (grant OPP1084307) to The INCLEN Trust International and sub-grant to Manipal University (subgrant INC2015GNT004)., during the conduct of the study and for the submitted work; no financial relationships with any organisations that might have an interest in the submitted work in the previous three years;no other relationships or activities that could appear to have influenced the submitted work.

**Patient consent** Not Applicable/ No human subjects.

**Provenance and peer review** Not commissioned; externally peer reviewed.

**Data sharing statement** All data supporting this study will be provided as supplementary material together with the manuscript of the study's final results.

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
