## [Reviewer comments · BMJ Open]

ARTICLE DETAILS

TITLE (PROVISIONAL)	Factors associated with mortality due to neonatal pneumonia in India: a protocol for systematic review and planned meta-analysis
AUTHORS	Nair, N.; Lewis, Leslie; Lakiang, Theophilus; Godinho, Myron; Murthy, Shruti; Venkatesh, Bhumika

VERSION 1 - REVIEW

REVIEWER	Dr Habib Hasan Farooqui Public Health Foundation of India
REVIEW RETURNED	18-May-2017

GENERAL COMMENTS	The authors have tried to address an important research gap in childhood pneumonia with appropriate research design. Few observations for clarification. Authors have mentioned conference papers in exclusion criteria, however section on hand searching mentions conference papers search for full paper and references. Although the study is based on secondary data, an IEC exemption may be considered. Finally, explicit definitions for outcomes and risk factors for neonatal pneumonia would strengthen the protocol.
--

REVIEWER	Andrew J Jones Paediatric and Neonatal Intensive Care Units Great Ormond Street Hospital London UK
REVIEW RETURNED	27-May-2017

GENERAL COMMENTS	This is an important research question and the authors present a study protocol which adheres to the PRISMA-P checklist. There are, however, some problems with the protocol which may prevent the research question being fully realised. - The term neonate is not defined. One presumes this includes newborns up to 28 days of age. The newborns presenting with pneumonia on day 1, day 7, and day 28 are going to be quite different in terms of risk factors, infecting organism and outcome. For example, in the first week of life gram negative organisms are more common, whereas gram positive organisms are more common in weeks 2,3 and 4. Combining all these neonates into one operational group may not be useful and may mask significant findings. The authors may wish to consider being more specific in their inclusion criteria. There is a risk of such heterogeneity of studies and outcomes that a meaningful meta-analysis is not
--

possible.

- The authors do not define neonatal pneumonia. This is an understandably pragmatic approach as the definition is likely to vary significantly from paper to paper, and there is no accepted, validated definition. I, however, foresee two problems. The study sets out to identify studies of neonates with pneumonia independent of sepsis, but I'm not sure this distinction can be made with any reliability. The WHO does not distinguish between neonatal pneumonia and other forms of severe sepsis because there is such overlap in clinical signs. I do not see any detriment to including studies looking at neonates with pneumonia AND sepsis, and the number of relevant papers may be increased. Secondly, definitions of neonatal pneumonia can include risk factors in their criteria - such as maternal fever or PROM - (Mathur NB, Garg K, Kumar S. Respiratory distress in neonates with special reference to pneumonia. Indian Pediatr 2002;39:529-37) and if these studies were to be included in the analysis the investigation of which risk factors are present in neonatal pneumonia would become a circular argument. This may need to be accounted for in terms of which studies are included.

- It is not entirely clear from the protocol how the results of the study will inform decisions about patient management. Is the meta-analysis to address patient factors (e.g. hypoxaemia at presentation) or population factors (e.g. private or public hospital)? Do the authors have an hypothesis about what factors might be important?

- The study is limited to Indian papers, and I understand the researchers are aiming to inform practice in India. But expanding the search to all developing countries, or at least to South Asia, may expand the pool of studies for inclusion, and will give the paper better generalisability, and more international appeal (I concede that the bulk of literature will likely end up being from India anyway).

- In order to make the data extraction more rigorous and objective, had the researchers considered blinding the two researchers to the paper details (i.e. obscuring the journal title, authors and institution)?

- The protocol could be more explicit about how the results will be presented and the sort of graphical displays that may be utilised (e.g. forest plot with logarithmic scale for ratios).

- The protocol states that this is part of a trilogy of meta-analyses, but I'm not sure how 'risk factors' and 'predictors of mortality' are different in this context; or at least they are not different enough to warrant separate studies. I wonder if the predictors of mortality will be the risk factors from this meta-analysis which show statistical significance.

- What is your timescale for mortality from pneumonia? Within the neonatal period, 90 days etc?

VERSION 1 – AUTHOR RESPONSE

Reviewer: 1

Dr Habib Hasan Farooqui

Public Health Foundation of India

Please state any competing interests or state 'None declared': None

Please leave your comments for the authors below

The authors have tried to address an important research gap in childhood pneumonia with appropriate research design. Few observations for clarification.

Authors have mentioned conference papers in exclusion criteria, however section on hand searching mentions conference papers search for full paper and references.

Response: Thank you for the observation. This line has been removed:

“(b) conference proceedings to review references and contact authors for full text of identified literature” (line nos. 51-52)

Although the study is based on secondary data, an IEC exemption may be considered.

Finally, explicit definitions for outcomes and risk factors for neonatal pneumonia would strengthen the protocol.

Response: As definitions for many of these medical entities are variable (including the definition of neonatal pneumonia itself), we have stipulated that “Definitions as reported by the authors will be considered and captured in the review” (line nos 34-35)

Reviewer: 2

Andrew J Jones

Paediatric and Neonatal Intensive Care Units, Great Ormond Street Hospital, London, UK

Please state any competing interests or state 'None declared': None declared

Please leave your comments for the authors below

This is an important research question and the authors present a study protocol which adheres to the PRISMA-P checklist. There are, however, some problems with the protocol which may prevent the research question being fully realised.

- The term neonate is not defined. One presumes this includes newborns up to 28 days of age.

Response: We noticed heterogeneity in the definitions of 'neonate' in the Indian context. Our intent, when we stated that “Definitions as reported by the authors will be considered and captured in the review” (line nos 34-35), was to place emphasis on this very heterogeneity despite the accepted WHO definition (upto 28 days of age). Highlighting these inconsistencies in the final 'policy brief' is one of the intentions of this project.

The newborns presenting with pneumonia on day 1, day 7, and day 28 are going to be quite different in terms of risk factors, infecting organism and outcome. For example, in the first week of life gram negative organisms are more common, whereas gram positive organisms are more common in weeks 2,3 and 4. Combining all these neonates into one operational group may not be useful and may mask significant findings. The authors may wish to consider being more specific in their inclusion criteria. There is a risk of such heterogeneity of studies and outcomes that a meaningful meta-analysis is not possible.

Response: We agree with the reviewer's comments. We have specified that we will perform a subgroup analysis by time of pneumonia onset as follows:

“Subgroup analysis will be done, when data are available, to compare risk factors according to, but not limited to (a) study design (b) type of neonatal pneumonia, (c) setting, and (d) timing of onset (early and late) of neonatal pneumonia.” (line nos. 50-55)

- The authors do not define neonatal pneumonia. This is an understandably pragmatic approach as the definition is likely to vary significantly from paper to paper, and there is no accepted, validated definition. I, however, foresee two problems. The study sets out to identify studies of neonates with pneumonia independent of sepsis, but I'm not sure this distinction can be made with any reliability. The WHO does not distinguish between neonatal pneumonia and other forms of severe sepsis because there is such overlap in clinical signs. I do not see any detriment to including studies looking at neonates with pneumonia AND sepsis, and the number of relevant papers may be increased.

Response: Thank you for your recommendation. We too deliberated regarding the same during the development of this protocol. However, as this project is part of a larger project on 'Childhood Pneumonia in India', and was funded for the express purpose of generating evidence on pneumonia among neonates in particular, we were required to restrict our scope to only include studies which either reported on neonatal pneumonia or the pneumonia component of neonatal sepsis. We developed a comprehensive search strategy, to ensure that all studies on neonatal sepsis could be captured during the search. Thus, during screening, sepsis studies which explicitly provided data on their pneumonia component would not be missed.

Secondly, definitions of neonatal pneumonia can include risk factors in their criteria - such as maternal fever or PROM - (Mathur NB, Garg K, Kumar S. Respiratory distress in neonates with special reference to pneumonia. Indian Pediatr 2002;39:529-37) and if these studies were to be included in the analysis the investigation of which risk factors are present in neonatal pneumonia would become a circular argument. This may need to be accounted for in terms of which studies are included.

Response: We will report all the risk factors as reported by the included studies as well as the definitions for neonatal pneumonia used to make the diagnosis in these studies. Should we find such studies, we will be sure to explicitly alert the reader to this issue as well to the implications of drawing conclusions from this data.

- It is not entirely clear from the protocol how the results of the study will inform decisions about patient management. Is the meta-analysis to address patient factors (e.g. hypoxaemia at presentation) or population factors (e.g. private or public hospital)? Do the authors have an hypothesis about what factors might be important?

Response: We anticipate that findings of this review will draw attention to research on the epidemiology of neonatal pneumonia and highlight areas that will require more attention from local researchers. Identifying relevant risk factors specific to the Indian context (socio-cultural, economic, etc) is another specific intention for conducting this review. The meta-analysis is aimed to address any factor (patient or population) for which quantitative data is currently available. We did not hypothesize as to which factors might be important in this context.

- The study is limited to Indian papers, and I understand the researchers are aiming to inform practice in India. But expanding the search to all developing countries, or at least to South Asia, may expand the pool of studies for inclusion, and will give the paper better generalisability, and more international appeal (I concede that the bulk of literature will likely end up being from India anyway).

Response: We recognize the importance of this suggestion and considered doing the same during our protocol development process. However, as this project is part of a larger project on 'Childhood Pneumonia in India', and was funded for the express purpose of generating evidence specific to the Indian context, we were required to restrict our scope to only include studies conducted in the Indian context.

- In order to make the data extraction more rigorous and objective, had the researchers considered blinding the two researchers to the paper details (i.e. obscuring the journal title, authors and

institution)?

Response: We did not blind researchers to the paper details specified above. However, we provided the research team with adequate training and conducted supervised pilot testing of the process to ensure objectivity and validity, to reduce bias.

- The protocol could be more explicit about how the results will be presented and the sort of graphical displays that may be utilised (e.g. forest plot with logarithmic scale for ratios).

Response: We have included the following in the text:

“illustrated in a forest plot (using a logarithmic scale to present ratios in case event data is not available for analysis) along with tables where necessary.¹⁶ We will summarize the characteristics and results of included studies using additional tables, supplemented by a narrative summary that will compare and evaluate methods used and principal results between studies. Further, we will describe factors for which a meta-analysis is impossible and the reasons for exclusion from meta-analysis will be provided.”

- The protocol states that this is part of a trilogy of meta-analyses, but I'm not sure how 'risk factors' and 'predictors of mortality' are different in this context; or at least they are not different enough to warrant separate studies. I wonder if the predictors of mortality will be the risk factors from this meta-analysis which show statistical significance.

Response: Thank you for this observation. We acknowledge that our statement may have been misleading and have amended it to read as follows, as this more accurately represents the objectives of the other systematic reviews in this series:

“This protocol is part of a larger mixed-methods research project consisting of a qualitative study and a trilogy of systematic reviews on neonatal pneumonia in India addressing

- factors associated with neonatal pneumonia,
- treatment options and barriers to the case management of neonatal pneumonia
- factors associated with mortality due to neonatal pneumonia.”

- What is your timescale for mortality from pneumonia? Within the neonatal period, 90 days etc?

Response: Mortality from neonatal pneumonia had to occur within the stipulated 'neonatal' period. We have included this line in the protocol.

VERSION 2 – REVIEW

REVIEWER	Habib Hasan Farooqui Public Health Foundation of India
REVIEW RETURNED	26-Jun-2017

GENERAL COMMENTS	The manuscript can be accepted.
---------------------------------